# Non-Deterministic Planning for Hyperproperty Verification

**Primary Keywords:** *None*

## Abstract

Non-deterministic planning aims to find a policy that achieves a given objective in an environment where actions have uncertain effects, and the agent – potentially – only observes parts of the current state. Hyperproperties are properties that relate multiple paths of a system and can, e.g., capture security and information-flow policies. Popular logics for expressing hyperproperties – such as HyperLTL – extend LTL by offering selective quantification over executions of a system. In this paper, we show that planning offers a powerful intermediate language for the automated verification of hyperproperties. Concretely, we present an algorithm that, given a HyperLTL verification problem, constructs a non-deterministic multi-agent planning instance (in the form of a QDec-POMDP) that, when admitting a plan, implies the satisfaction of the verification problem. We show that for large fragments of HyperLTL, the resulting planning instance corresponds to a classical, FOND, or POND planning problem. We implement our encoding in a prototype verification tool and report on encouraging experimental results using off-the-shelf FOND planners.

## 1 Introduction

AI planning is the task of finding a policy (aka. plan) that ensures that a specified goal is reached. In this paper, we present an exciting new application of planning: the automated verification of *hyperproperties*.

**Hyperproperties and HyperLTL.** Hyperproperties generalize traditional trace properties by relating *multiple* executions of a system (Clarkson and Schneider 2008). A trace property – specified, e.g., in LTL – reasons about *individual* executions in isolation, which falls short in many applications. For example, assume we want to specify that the output of a system (modeled via atomic proposition $o$) only depends on some low-security input $l$ and does not leak information about a secret input $h$. We cannot specify this as a trace property in LTL; we need to relate multiple executions to observe how different inputs impact the output. HyperLTL extends LTL with explicit quantification over executions (Clarkson et al. 2014), and allows for the specification of such a property. For example, we can express observational determinism (Zdancewic and Myers 2003) as follows:

$$\forall \pi.\, \forall \pi'.\, (l_\pi \leftrightarrow l_{\pi'}) \rightarrow \mathsf{G}(o_\pi \leftrightarrow o_{\pi'}) \qquad \text{(OD)}$$

This formula states that on any *pair* of traces $\pi, \pi'$ with identical low-security input, the output is (globally) the same. In other words, the system is deterministic in the low-security inputs. For non-deterministic systems, (OD) is often too strict, as any given low-security input might lead to multiple outputs. A relaxed notation – called non-inference (NI) (McLean 1994) – can be expressed in HyperLTL as follows:

$$\forall \pi.\, \exists \pi'.\, \mathsf{G}\left((o_\pi \leftrightarrow o_{\pi'}) \wedge (l_\pi \leftrightarrow l_{\pi'}) \wedge \neg h_{\pi'}\right) \qquad \text{(NI)}$$

That is, for any execution $\pi$, there *exists* another execution $\pi'$ that has the same low-security behavior (via propositions $o, l$), but yet has a fixed "dummy" high-security input (in our case, we require that $h$ is always false, i.e., never holds on $\pi'$). If NI holds, a low-security attacker can not distinguish any high-security input from the dummy input.

**HyperLTL Verification as Planning.** Our goal is to automatically verify that a finite-state system $\mathcal{T}$ satisfies a HyperLTL formula $\varphi$. We introduce a novel verification approach that leverages the advanced methods developed within the planning community. Concretely, we present a reduction that soundly converts a HyperLTL verification problem into a planning problem. Depending on the HyperLTL formula, our encoding uses several advanced features supported by modern planning frameworks, such as uncertain action effects (non-determinism) (Cimatti et al. 2003), partial observations (Bertoli et al. 2006), and multiple agents. We show that – by carefully combining these features – we obtain a planning problem that is sound w.r.t. the HyperLTL semantics: every plan can be translated back into a validity witness for the original verification problem. As a consequence, our encoding allows us to utilize mature planning tools for the verification of complex HyperLTL properties. We implement our encoding as a prototype and demonstrate that existing off-the-shelf planners outperform previous game-based verification methods for HyperLTL.

## 2 High-Level Overview

Before proceeding with a formal construction, we provide some high-level intuition of our encoding. In HyperLTL, we can quantify over the executions of a system (as seen informally in OD and NI). The overarching idea in our encoding is to let the planning agent control all *existentially quantified* executions, such that any valid plan directly corresponds to a witness for the existentially quantified executions.

**Verification as Planning.** As an example, assume we want to verify that (OD) does *not* hold on a given system $\mathcal{T}$, i.e., we want to find concrete executions $\pi, \pi'$ that violate the body of (OD). We can interpret this as a classical (single-agent) planning problem: each planning state maintains two locations in $\mathcal{T}$, one for $\pi$ and one for $\pi'$, and, in each step, the actions update the locations for $\pi, \pi'$ by moving along the transitions in $\mathcal{T}$. The planning objective is to construct executions for $\pi, \pi'$ that *violate* $(l_\pi \leftrightarrow l_{\pi'}) \rightarrow \mathsf{G}(o_\pi \leftrightarrow o_{\pi'})$. Any successful plan (i.e., sequence of transitions) then directly corresponds to concrete paths $\pi, \pi'$ disproving (OD).

**Verification as Non-deterministic Planning.** Verification becomes more interesting when the HyperLTL formula contains alternations, as in (NI). Following the above intuition, a plan should provide a concrete witness for (the existentially quantified) $\pi'$, but – this time – we need to consider *all* executions for (the universally quantified) $\pi$. Our idea is that we can approximate this behavior by viewing it as a fully-observable non-deterministic (FOND) planning problem; intuitively, a plan controls the behavior of $\pi'$ while the behavior of $\pi$ is non-deterministic. That is, each action determines a successor location for $\pi'$ but also non-deterministically updates the location of $\pi$. The agent's object is to ensure that $\pi, \pi'$ satisfy $\mathsf{G}\big((o_\pi \leftrightarrow o_{\pi'}) \wedge (l_\pi \leftrightarrow l_{\pi'}) \wedge \neg h_{\pi'}\big)$ (the body of NI). Any plan (which is now *conditional* on the non-deterministic outcomes) thus defines a concrete execution for $\pi'$, depending on the concrete execution for $\pi$.

**Verification as Planning Under Partial Observations.** In (NI), $\pi'$ is quantified *after* $\pi$, so the action sequence that defines the behavior of $\pi'$ can be based on the behavior of $\pi$. This changes when quantifiers *succeed* existential quantification, e.g., in a formula of the form $\exists \pi. \forall \pi'$. For such formulas, we follow the same idea as before but ensure that the actions controlling $\pi$ are *independent* of the current state of $\pi'$, i.e., the agent must act under *partial information*.

This intuition generalizes to *full* HyperLTL by introducing one agent for each existential quantifier and carefully designing the observations of each agent (cf. Section 6).

## 3  Related Work

Non-deterministic planning provides a powerful intermediate language that encompasses problems such as reactive synthesis (Camacho et al. 2018), controller synthesis in MDPs, epistemic planning (Engesser and Miller 2020), and generalized planning (Hu and Giacomo 2011). Consequently, many methods and tools have been developed (Pereira et al. 2022; Messa and Pereira 2023; Camacho et al. 2017; Mokhtari et al. 2021; Geffner and Geffner 2018; Rodriguez et al. 2021; Muise, McIlraith, and Beck 2012; Kuter et al. 2008), with some also supporting partial observations (Bertoli et al. 2006; Cimatti et al. 2003; Bonet and Geffner 2011). In terms of HyperLTL verification, complete verification is possible via expensive automata complementations (Finkbeiner, Rabe, and Sánchez 2015), or cheaper (but incomplete) bounded methods (Hsu, Sánchez, and Bonakdarpour 2021). For $\forall^* \exists^*$ HyperLTL properties, our encoding is related to the parity-game-based approach of Beutner

and Finkbeiner (2022), where one player controls existentially quantified executions. Crucially, the size of their game scales *exponentially* in the number of quantified executions, making it impractical for larger instances. In contrast, the planning-based approach in this paper can describe the problem compactly (locally) and let the planner determine how to best explore the state space. Our experimental results show that this leads to large performance gains in practice (cf. Section 7). Moreover, our planning-based encoding is applicable to arbitrary quantifier prefixes and thus provides a verification method for the full logic, not only $\forall^* \exists^*$ formulas.

## 4  Planning Preliminaries

As a basic planning model, we use Qualitative Dec-POMDP (QDec-POMDP), a general model that encompasses multiple agents, non-deterministic effects, and partial observations (Brafman, Shani, and Zilberstein 2013).

**Definition 1.** *A QDec-POMDP is a tuple* $\mathcal{G} = (I, S, S_0, \{A_i\}, \delta, \{\Omega_i\}, \{\omega_i\}, G)$, *where* $I = \{1, \ldots, m\}$ *is a finite set of agents;* $S$ *is a finite set of states; and* $S_0 \subseteq S$ *is a set of initial states; For each* $i \in I$, $A_i$ *is a finite set of actions and we define* $\vec{A} = \otimes_{i \in I} A_i$ *as the set of joint actions.* $\delta : S \times \vec{A} \rightarrow 2^S$ *is a (non-deterministic) transition function; For each* $i \in I$, $\Omega_i$ *is a finite set of observations, and the observation function* $\omega_i : S \rightarrow \Omega_i$ *gives* $i$'s *local observation; Lastly,* $G \subseteq S$ *is a set of goal states.*

We write $\{a_i\} \in \vec{A}$ for the joint action where each agent $i \in I$ chooses action $a_i$. A *local policy* for an agent $i \in I$, is a conditional plan that picks an action based on the history of observations, i.e., a function $f_i : \Omega_i^+ \rightarrow A_i$ (represented, e.g., as a tree of degree $|\Omega_i|$ where nodes are labeled with elements from $A_i$). A *joint policy* $\{f_i\}$ assigns each agent $i \in I$ a local policy $f_i$. A finite path $p \in S^+$ is *compatible with* $\{f_i\}$ if and only if **(1)** $p(0) \in S_0$ (i.e., the path starts in an initial state), and **(2)** for every $0 \leq k < |p|$, $p(k+1) \in \delta(p(k), \{a_i\})$ where $a_i = f_i(\omega_i(p(0)) \cdots \omega_i(p(k)))$. That is, in every step, we compute the joint action $\{a_i\}$, where each $a_i$ is determined by policy $f_i$ based on the past observations made by $i$ on the prefix $p(0) \cdots p(k)$. We write $Exec(\{f_i\}) \subseteq S^+$ for the set of all $\{f_i\}$-compatible paths.

The objective of the agents is to reach a goal state in $G$. Following Cimatti et al. (2003), we distinguish between different levels of reachability. A policy is a *weak plan* if *some* $p \in Exec(\{f_i\})$ reaches a state in $G$, i.e., $\{f_i\}$ can reach the goal provided the non-determinism is resolved favorably. A policy is a *strong plan* if there exists a $N \in \mathbb{N}$ such that *every* $p \in Exec(\{f_i\})$ with $|p| \geq N$ reaches $G$, i.e., the goal is guaranteed to be reached, irrespective of non-deterministic outcomes. Finally, a policy is a *strong cyclic plan* if, for every $p \in Exec(\{f_i\})$, either $p$ reaches $G$ or there exists some $p' \in Exec(\{f_i\})$ that extends $p$ (i.e., $p$ is a *prefix* of $p'$) and reaches $G$.[1]

---

[1] A strong cyclic plan is one that always *preserves the possibility of reaching the goal*, i.e., at every point, the non-determinism can be resolved favorably such that the goal is reached. Our definition expresses exactly this: either $p$ already reaches $G$ or some extension of $p$ *can* reach the goal. This definition is equivalent to the one of Cimatti et al. (2003).

## 5 Hyperproperties and HyperLTL

We assume that $AP$ is a fixed set of *atomic propositions*.

**Transition Systems.** As the basic system model, we use finite-state transition systems (TSs), which are tuples $\mathcal{T} = (L, l_{init}, \mathbb{D}, \kappa, \ell)$ where $L$ is a finite set of locations (we use "locations" to distinguish them from the "states" in a planning domain), $l_{init} \in L$ is an initial location, $\mathbb{D}$ is a finite set of *directions*, $\kappa : L \times \mathbb{D} \to L$ is the transition function, and $\ell : L \to 2^{AP}$ labels each location with an evaluation of the APs. We use explicit directions in order to uniquely identify successor locations; we can easily model a traditional transition function $\kappa : L \to 2^L \setminus \{\emptyset\}$ using directions. A path in $\mathcal{T}$ is an infinite sequence $p \in L^\omega$ such that $p(0) = l_{init}$, and for every $k \in \mathbb{N}$, there exists some $d \in \mathbb{D}$ s.t. $p(k+1) = \kappa(p(k), d)$. We define $Paths(\mathcal{T}) \subseteq L^\omega$ as the set of all paths in $\mathcal{T}$.

**HyperLTL.** As the basic specification language for hyperproperties, we use HyperLTL, an extension of LTL with explicit quantification over (execution) paths (Clarkson et al. 2014). Let $\mathcal{V} = \{\pi, \pi', \dots\}$ be a set of *path variables*. HyperLTL formulas are generated by the following grammar

$$\psi := a_\pi \mid \psi \wedge \psi \mid \neg\psi \mid \psi \, \mathsf{U} \, \psi \mid \mathsf{X} \, \psi$$
$$\varphi := \mathbb{Q}\pi. \, \varphi \mid \psi$$

where $a \in AP$, $\pi \in \mathcal{V}$, and $\mathbb{Q} \in \{\forall, \exists\}$. We use the usual derived boolean and temporal constants and operators $true, false, \vee, \to, \leftrightarrow, \mathsf{F} \, \psi := true \, \mathsf{U} \, \psi, \mathsf{G} \, \psi := \neg \mathsf{F} \, \neg\psi$.

Given a TS $\mathcal{T} = (L, l_{init}, \mathbb{D}, \kappa, \ell)$, we evaluate a HyperLTL formula in the context of a path assignment $\Pi : \mathcal{V} \rightharpoonup L^\omega$ (mapping path variables to paths) as follows:

$$\Pi, i \models_\mathcal{T} a_\pi \quad \text{iff} \quad a \in \ell\big(\Pi(\pi)(i)\big)$$
$$\Pi, i \models_\mathcal{T} \psi_1 \wedge \psi_2 \quad \text{iff} \quad \Pi, i \models_\mathcal{T} \psi_1 \text{ and } \Pi, i \models_\mathcal{T} \psi_2$$
$$\Pi, i \models_\mathcal{T} \neg\psi \quad \text{iff} \quad \Pi, i \not\models_\mathcal{T} \psi$$
$$\Pi, i \models_\mathcal{T} \mathsf{X}\psi \quad \text{iff} \quad \Pi, i+1 \models_\mathcal{T} \psi$$
$$\Pi, i \models_\mathcal{T} \psi_1 \mathsf{U} \psi_2 \quad \text{iff} \quad \exists j \geq i. \, \Pi, j \models_\mathcal{T} \psi_2 \text{ and}$$
$$\forall i \leq k < j. \, \Pi, k \models_\mathcal{T} \psi_1$$
$$\Pi, i \models_\mathcal{T} \mathbb{Q}\pi. \varphi \quad \text{iff} \quad \mathbb{Q}p \in Paths(\mathcal{T}). \, \Pi[\pi \mapsto p], i \models_\mathcal{T} \varphi$$

The atomic formula $a_\pi$ holds whenever $a$ holds in the current position $i$ on the path bound to $\pi$ (as given by $\ell$). Boolean and temporal operators are evaluated as expected by updating the current evaluation position $i$, and quantification adds paths to $\Pi$. We refer to Finkbeiner (2023) for details. We say $\mathcal{T}$ satisfies $\varphi$, written $\mathcal{T} \models \varphi$, if $\{\}, 0 \models_\mathcal{T} \varphi$, where $\{\}$ denotes the path assignment with empty domain.

## 6 Verification via Planning

We want to automatically verify that $\mathcal{T} \models \varphi$. To this end, we present a novel encoding into a planning problem, thus leveraging the extensive research and tool development within the planning community. As already outlined in Section 2, our main idea is to interpret existential quantification in $\varphi$ as being resolved by an agent that picks transitions in $\mathcal{T}$ to construct a path. Throughout this section, we assume that $\mathcal{T} = (L, l_{init}, \mathbb{D}, \kappa, \ell)$ is the fixed TS and $\varphi = \mathbb{Q}_1\pi_1 \dots \mathbb{Q}_n\pi_n. \psi$ the fixed HyperLTL formula over path variables $\pi_1, \dots, \pi_n$.

**DFAs and Reachability Specifications.** A deterministic finite automaton (DFA) over some alphabet $\Sigma$ is a tuple $\mathcal{A} = (Q, q_0, \varrho, F)$ where $Q$ is a finite set of states, $q_0 \in Q$ is an initial state, $\varrho : Q \times \Sigma \to Q$ is a deterministic transition function, and $F \subseteq Q$ is a set of accepting states. An infinite word $u \in \Sigma^\omega$ is accepted by $\mathcal{A}$ if the unique run *eventually* reaches some state in $F$. We say $\varphi$ is a *Reachability HyperLTL formula* if $\psi$ (the LTL-like body of $\varphi$) is recognized by a DFA, i.e., some DFA $\mathcal{A}_\psi = (Q_\psi, q_{0,\psi}, \varrho_\psi, F_\psi)$ over alphabet $2^{AP \times \{\pi_1, \dots, \pi_n\}}$ accepts exactly those infinite words that satisfy $\psi$ (recall that the atoms in the LTL-like formula $\psi$ have the form $a_{\pi_i} \in AP \times \{\pi_1, \dots, \pi_n\}$).

### 6.1 Encoding as a Planning Problem

We write $\mathcal{V}_\exists = \{\pi_i \mid \mathbb{Q}_i = \exists\}$ for existentially quantified path variables in $\varphi$, and $\mathcal{V}_\forall$ for universally quantified ones.

**Definition 2.** *Define* $\mathcal{G}_{\mathcal{T},\varphi}^{reach} := (I, S, S_0, \{A_i\}, \delta, \{\Omega_i\}, \{\omega_i\}, G)$ *where* $I := \{i \mid \pi_i \in \mathcal{V}_\exists\}$; $S := \{\langle l_1, \dots, l_n, q\rangle \mid q \in Q_\psi \wedge \forall j. l_j \in L\}$; *and* $S_0 := \{\langle l_{init}, \dots, l_{init}, q_{0,\psi}\rangle\}$. *For each* $i \in I$, *we set* $A_i := \mathbb{D}$ *and define the transitions by*

$$\delta(\langle l_1, \dots, l_n, q\rangle, (\{d_j\}_{\pi_j \in \mathcal{V}_\exists})) :=$$
$$\Big\{ \langle \kappa(l_1, d_1), \dots, \kappa(l_n, d_n), q'\rangle \mid \forall \pi_j \in \mathcal{V}_\forall. \, d_j \in \mathbb{D} \wedge$$
$$q' = \varrho_\psi(q, \bigcup_{j=1}^n \{(a, \pi_j) \mid a \in \ell(l_j)\}) \Big\},$$

$\Omega_i := \{\langle l_1, \dots, l_i\rangle \mid \forall j \leq i. \, l_j \in L\}$; $\omega_i(\langle l_1, \dots, l_n, q\rangle) := \langle l_1, \dots, l_i\rangle$; *and* $G := \{\langle l_1, \dots, l_n, q\rangle \mid q \in F_\psi\}$.

Let us step through this definition step-by-step. As already hinted in Section 2, we add one agent $i$ for each existentially qualified path $\pi_i \in \mathcal{V}_\exists$. Each state has the form $\langle l_1, \dots, l_n, q\rangle$ and tracks a current location for each of the paths (where $l_j \in L$ is the current location for path $\pi_j$), and $q$ tracks the current state of $\mathcal{A}_\psi$. Intuitively, the planning problem simulates $\pi_1, \dots, \pi_n$ by keeping track of their current location $(l_1, \dots, l_n)$, and letting the actions chosen by the agents (for existentially quantified paths) or the nondeterminism (for universally quantified paths) fix the next location. We start each $\pi_j$ in the initial location $l_{init}$ and start the run of $\mathcal{A}_\psi$ in the initial state $q_{0,\psi}$. The actions of each agent then directly correspond to directions in $\mathcal{T}$. When given a joint action $\{d_j\}_{\pi_j \in \mathcal{V}_\exists}$ (i.e., a direction for each existentially quantified path), the transition formula considers all possible directions for universally quantified paths and updates each location $l_j$ based on the direction $d_j$. Existentially quantified paths thus follow the direction selected by the respective agent, and universally quantified ones follow a non-deterministically chosen direction. In each step, we also update the state of $\mathcal{A}_\psi$: For each $1 \leq j \leq n$, we collect all APs that hold in the current location $\ell(l_j)$ and index them with $\pi_j$, thus obtaining a letter in $2^{AP \times \{\pi_1, \dots, \pi_n\}}$ which we feed to the transition function of $\mathcal{A}_\psi$. As argued in Section 2, each agent $i$ controlling $\pi_i \in \mathcal{V}_\exists$ may only observe the traces $\pi_1, \dots, \pi_i$ quantified *before* $\pi_i$, so the observation set $\Omega_i$ of agent $i$ consist exactly of states of the form $\langle l_1, \dots, l_i\rangle$ and the observation function $\omega_i$ projects each state to the observable locations. Lastly, the goal consists of all states in which the automaton has reached one of $\mathcal{A}_\psi$'s accepting states.

**Theorem 1.** *Assume $\varphi$ is a reachability HyperLTL formula. If $\mathcal{G}_{\mathcal{T},\varphi}^{reach}$ admits a strong plan, then $\mathcal{T} \models \varphi$.*

*Proof Sketch.* We can use the policies in a strong plan for $\mathcal{G}_{\mathcal{T},\varphi}^{reach}$ to construct *Skolem functions* for existentially quantified paths in $\varphi$. The full proof is complex and provided in the supplementary materials. □

**Factored Representation.** In Definition 2, we used an explicit-state (flat) representation of the problem with $|L|^n * |Q_\psi|$ states. In practice, many planning formats (e.g., STRIPS, PDDL, SAS) allow for a *factored* description of the state space, using roughly $n \cdot |L| + |Q_\psi|$ many fluents that track the current location of each path *individually*. The possibility of using a factored representation is a core motivation for using planning tools for HyperLTL verification. In Section 7, we will show that this factored representation also leads to performance improvements over the SOTA.

### 6.2 Encoding for Safety Properties

In the construction above, we assumed the $\varphi$ denotes a reachability property. We can also handle the case in which $\varphi$ denotes a *safety HyperLTL formula*, i.e., $\psi$ expresses that "something bad may never happen". In the safety case, we again model $\psi$ as a DFA $\mathcal{A}_\psi = (Q_\psi, q_{0,\psi}, \varrho_\psi, F_\psi)$, but now say that an infinite word is accepted if it *never* visits a state in $F_\psi$. As an example, (NI) is a safety HyperLTL formula.

Different from reachability properties, safety properties reason about infinite executions (and not only finite prefixes thereof), so phasing it as a planning problem requires modifications. First, we add special sink states $s_{win}$ and $s_{lose}$, and mark $s_{win}$ as the unique goal state. From any state $\langle l_1, \ldots, l_n, q \rangle$ where $q \in F_\psi$, we then deterministically move to $s_{lose}$. Conversely, from any state $\langle l_1, \ldots, l_n, q \rangle$ where $q \notin F_\psi$, we extend the transitions in Definition 2 with an additional non-deterministic transition to $s_{win}$. The agents can thus never ensure a visit to $s_{win}$, but a strong *cyclic* plan guarantees that we never visit a state in $F_\psi$. We denote the resulting QDec-POMPD with $\mathcal{G}_{\mathcal{T},\varphi}^{safe}$; a full description can be found in the supplementary materials.

**Theorem 2.** *Assume $\varphi$ is a safety HyperLTL property. If $\mathcal{G}_{\mathcal{T},\varphi}^{safe}$ admits a strong cyclic plan, then $\mathcal{T} \models \varphi$.*

**Full HyperLTL.** Our construction can also be extended to handle *full* HyperLTL by reducing to planning problems with temporal goals specified in LTL(f) (Patrizi et al. 2011; Camacho et al. 2017; Camacho and McIlraith 2019). We restrict our constriction to the case of reachability and safety properties as (1) this suffices for almost all properties of interest, and (2) it allows us to employ automated planners that yield strong (cyclic) plans for non-temporal objectives.

### 6.3 Easier Planning Problems

In general, our encoding yields a planning problem that combines multiple agents, non-determinism, and partial observations. In many situations, however, the resulting problem does not require all these features: (1) For $\exists^*$ properties, the planning problem is classical, i.e., consists of a single agent, deterministic actions, and full information. (2)

| Model | Size PG | Size PDDL | $\exists\exists$ | | $\forall\exists$ | |
| --- | --- | --- | --- | --- | --- | --- |
| | | | $t_{PG}$ | $t_{\text{HyPlan}}$ | $t_{PG}$ | $t_{\text{HyPlan}}$ |
| BAKERY3 | 31016.4 | 16.0/8.0/75.7 | 11.9 | **1.21** | 13.2 | **0.96** |
| BAKERY5 | 614.6 | 16.2/7.7/19.0 | 2.10 | **0.68** | 3.52 | **0.73** |
| MUTATION | 1807.5 | 16.7/78.5/9.8 | 4.26 | **0.72** | 6.75 | **0.43** |
| NI_C | 1370.3 | 15.2/7.7/13.5 | 3.40 | **0.42** | 4.41 | **0.41** |
| NI_I | 948.3 | 17.0/8.5/104.3 | **4.22** | 4.31 | 5.56 | **5.11** |
| NRP_C | 1688.3 | 15.8/7.7/23.0 | 6.89 | **0.48** | 8.37 | **0.75** |
| NRP_I | 1018.6 | 15.8/7.7/22.2 | 7.21 | **0.36** | 7.45 | **0.47** |
| SNARK_CON | 105854.7 | 15.2/7.7/192.1 | 27.02 | **7.15** | 39.82 | **8.17** |
| SNARK_SEQ | 17415.6 | 16.0/8.0/32.4 | 13.66 | **0.65** | 11.56 | **5.15** |

Table 1: We compare `HyPlan` with a parity-game-based encoding on $\exists\exists$ and $\forall\exists$ properties. We list the time for both tools in seconds (averaged over 10 random formulas). Additionally, we give the average size of the PG, and the number of predicates/actions/objects in the PDDL encoding.

For $\forall^*\exists^*$ properties (e.g., NI), the problem involves a single agent acting under full information (FOND-planning). (3) For $\forall^*\exists^*\forall^*$ properties, the problem involves a single agent acting under partial observations (POND-planning).

## 7 Implementation and Experiments

We have implemented our encoding for $\forall^*\exists^*$ HyperLTL formulas in a prototype called `HyPlan`. Our tool produces FOND planning instance in an extension of PDDL, featuring `(oneof p1 ... pn)` expressions in action effects; A format widely supported by many FOND planners.

We compare `HyPlan` against the parity-game (PG) based encoding for $\forall^*\exists^*$ properties (Beutner and Finkbeiner 2022). For our experiments, we collect the 10 NuSMV models from Hsu, Sánchez, and Bonakdarpour (2021) and generate random formulas of the form $\exists\pi. \exists\pi'. \mathsf{F}\,\psi$ and $\forall\pi. \exists\pi'. \mathsf{G}\,\psi$ where $\psi$ is a temporal-operator-free formula. As remarked in Section 6.3, for the $\exists\exists$ properties `HyPlan` produces classical planning problems (which we solve using `Scorpion` (Seipp and Helmert 2018)), whereas the $\forall\exists$ properties yield FOND planning problems (which we solve using the FOND planner `MyND` (Mattmüller et al. 2010)). We report the average size of the PG and PDDL planning problem, as well as the time taken for the $\exists\exists$ and $\forall\exists$ properties in Table 1. As remarked in Section 3, the size of the PG is *exponential*, whereas the PDDL description is small and leaves the exact exploration strategy to the planner. Consequently, we observe that an existing (off-the-shelf) solver easily outperforms the game-based approach.

## 8 Conclusion

We have presented a novel application of non-deterministic planning: the verification of hyperproperties. Our encoding is applicable to formulas with *arbitrary* quantifier prefixes (beyond $\forall^*\exists^*$) and often yields classical or FOND planning instances that can be handled by existing mature planners. Our preliminary experiments show that off-the-shelf planners constitute an efficient verification method that outperforms existing game-based approaches. Moreover, any further progress into the development of non-deterministic planners (for which our work provides even more incentive) will directly improve our verification pipeline.

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
