# OpenReview forum: "Non-Deterministic Planning for Hyperproperty Verification"
_icaps-conference.org/ICAPS/2024/Conference — ICAPS 2024_

### Official Review · Reviewer_nsWc · 2024-01-07

**Significance And Importance:** 2
**Soundness:** 3
**Novelty:** 2
**Clarity:** 3
**Overall Evaluation:** 1
**Confidence:** 3

**Weaknesses:**

0: Minor weaknesses requiring some work to be addressed for the paper to be accepted.

**Contributions Of The Paper:**

This paper proposes a novel method for verification of hyperproperties expressed in HyperLTL by reducing the HyperLTL verification problem to a planning problem. The plan of planning problem is a witness for verification problem. The experimental result shows that the performance of the method proposed in this paper is superior to previous game-based verification methods.

**Ethical Considerations:**

(5) Excellent: The paper comprehensively addresses all of the applicable ethical considerations

**Nomination For Best Paper:**

No

**Questions For Authors:**

1. Can any hyperproperty verification problem instance be reduced to a planning problem instance?
2. Is there a hyperproperty in your experiment that cannot be verified successfully? If it exists, what is the reason for it?

**Reproducibility:**

5: Code and domains (whichever apply) are already publicly available

**Strengths Of The Paper:**

1. This paper proposes a novel method for verification of hyperproperties. This method can solve more hyperproperties verification problems expressed in HyperLTL formula and has better performance.
2. The definition paper is clear.

**Weaknesses Of The Paper:**

1. This paper is not well organized. I think the author should place the related work in the Introduction and place the details of hyperproperty and HyperLTL in the following section.
2. This paper lacks an example about encoding a HyperLTL formula into a planning problem.

Minor issues:
1.	In Introduction, “non-inference(NI)” => “non-interference(NI)”.
2.	In Section 6, “the fixed HyperLTL” => “is the fixed HyperLTL”

---

> ### Author Rebuttal · Authors · 2024-01-27
>
> We thank the reviewer for their positive review and valuable feedback.
> We will carefully consider the reviewer's suggestion regarding the structure when preparing a final version.
> If space permits, we will include a small example to illustrate our approach even better.
>
>
> **Questions:**
>
> 1) Our encoding allows us to reduce _arbitrary_ Reachability or Safety HyperLTL formulas into QDec-POMDPs.
> If we include temporal objectives, we can extend our construction to _full_ HyperLTL (as argued at the end of Section 6.2).
> Of course, it can be the case that the computed QDec-POMDP does not admit a plan.
>
> 2) In the experiments we tested, we could verify all hyperproperties.
> However, given the high computational complexity of HyperLTL, HyPlan cannot verify arbitrary instances; if the system grows too large, no planning tool is able to solve the resulting FOND planning instance. Nevertheless, our experiments in Table 1 indicate that our planning-based encoding scales to instances larger than those supported by a parity-game-based approach.

---

### Official Review · Reviewer_kpeV · 2024-01-21

**Significance And Importance:** 3
**Soundness:** 4
**Novelty:** 3
**Clarity:** 4
**Overall Evaluation:** 2
**Confidence:** 4

**Weaknesses:**

2: No major or minor weaknesses.

**Contributions Of The Paper:**

The paper introduces a novel application of AI Planning for the verification of hyperproperties expressed in HyperLTL. The approach developed by the authors is called HyPlan.The authors develop an encoding that can express hyperproperties (in HyperLTL)) using three different planning formalisms with different outcomes and assumptions: Classical Planning (Verification as (Deterministic) Planning), FOND Planning (Verification as Non-deterministic Planning), and POND Planning (Verification as Planning Under Partial Observation).
An empirical evaluation over 10 models shows the efficiency of HyPlan for hyperproperty verification, and it shows that HyPlan outperforms the parity-game-based encoding of Beutner and Finkbeiner (2022) in most of the used models.

**Ethical Considerations:**

(1) Not Applicable: The paper does not have any ethical considerations to address

**Nomination For Best Paper:**

No

**Questions For Authors:**

Please, address the following question in your response/rebuttal.

Q1. Why have you chosen to use MyND for non-deterministic planning?
Is that because MyND can extract both strong and strong-cyclic plans?
Is that because MyND can support both FOND and POND?

**Reproducibility:**

4: Authors promise to release code and domains (whichever apply).

**Strengths Of The Paper:**

The paper is well-written, organised, and easy to follow. The authors have done a good job at presenting the proposed ideas in a short paper (4 pages).
It is really nice to see the application of AI Planning techniques in other fields of study, in this case, the verification of hyperproperties.

The key strengths of the paper are as follows:
- A novel planning approach (HyPlan) for the verification of hyperproperties expressed in HyperLTL;
- The (hyperproperties) encoding as a planning problem, which is able to express hyperproperties in HyperLTL (for verification) using three different planning formalisms: Classical Planning (Verification as (Deterministic) Planning), FOND Planning (Verification as Non-deterministic Planning), and POND Planning (Verification as Planning Under Partial Observation).
- An empirical evaluation over 10 models comparing HyPlan against an existing parity-game-based encoding (Beutner and Finkbeiner 2022).

**Weaknesses Of The Paper:**

I see no weakness in the paper, but I do have some minor comments and points for consideration, as follows.

# Minor Comments and Points of Consideration:

C1. Does OD stand for Observational Determinism? Please, state that as you do for NI (Non-Inference).

C2. To some extent, your definition of a strong cyclic plan (Footnote 1) takes into account fairness, right? Please, clarify that.

C3. What does $L^{\omega}$ mean? Is that the set of possible paths?! Please, clarify that.

C4. In the Proof Sketch of Theorem 1, you say: "We can use the policies of a strong plan ...". Shouldn't it be "We can use the paths of a strong plan ...". Please, clarify that.

C5. SOTA stands for state-of-the-art, right?! Please, state that.

C6. You could extend your construction and representation to support full HyperLTL with temporally extended goals expressed in LTL(f) using FOND4LTLf (https://github.com/whitemech/FOND4LTLf).
That would be a nice extension and addition as future work for HyPlan.

# Writing Issues and Typos:

W1. "The agent’s object ..." -> "The agent’s objective ...".
W2. "We restrict our constriction" -> "construction"?!

---

> ### Author Rebuttal · Authors · 2024-01-27
>
> We thank the reviewer for their very positive review and valuable feedback.
> We will carefully take the reviewer's suggestions into account when preparing a final version.
>
> **Questions:**
>
> 1) We chose the MyND planner because it was easy to build on modern hardware (as it is written in Java) and supports both strong and strong-cyclic plans.
> During our experiments, this allowed us to employ the _same_ solver on all planning instances produced by HyPlan.
> As we use a standard extension of PDDL (see Section 7), HyPlan can be combined with any FOND planer.

---

### Official Review · Reviewer_s5CD · 2024-01-22

**Significance And Importance:** 2
**Soundness:** 3
**Novelty:** 3
**Clarity:** 4
**Overall Evaluation:** 2
**Confidence:** 4

**Weaknesses:**

0: Minor weaknesses requiring some work to be addressed for the paper to be accepted.

**Contributions Of The Paper:**

The paper presents a reduction for HyperLTL formulae towards a non-deterministic planning problem. The approach promises to be less expensive than the parity-game based approach of "Prophecy variables for Hyperproperty Verification" (2002).

**Ethical Considerations:**

(1) Not Applicable: The paper does not have any ethical considerations to address

**Nomination For Best Paper:**

No

**Questions For Authors:**

1) Is the DFA of section 6 a Büchi automaton? Looks like one.
2) In sec. 6.1, it's not clear how the non-determinism is treated. Does the planner selects a single outcome or does it progresses the state in a belief state (a set of possible destination states), keeping track of the whole uncertainty gained by the non-deterministic action? If a single outcome is selected, how this is done? My understanding is that, even if we're in a FOND, doing model-checking requires checking all the outcomes, i.e. a strong plan in Cimatti's sense.
3) I understand that formulae for checking deadlocks (as in invalid end states), correctness of system invariants, unreachability code,  and liveness properties through non-progress cycles (livelocks) are considered by the authors. What kind of expressvity do you expect from the QDec-POMDP translation?

**Reproducibility:**

3: Authors describe the implementation and domains in sufficient detail.

**Strengths Of The Paper:**

- The authors present a novel application for FOND planners: the verification of HyperLTL formulae.
- The "verification as planning" approach described is sound.
- The paper is well written and the approach is exhaustively presented, the Technical Background section is complete also in view of the brevity of the paper.
- A preliminary experimentation is presented, illustrating the feasibility of the approach.

**Weaknesses Of The Paper:**

Two aspects of the document seem particularly weak. First, the state of the art is well covered, but a thorough comparison with model checking approaches is necessary to convince the reader of the effectiveness of the approach. Second, experiments have been shown in the paper (which is remarkable, given that it's a short paper), but no comparison has been made with other algorithms. In comparison to [1], the extension of the experimental section is too small to evaluate the approach against other algorithms and the behaviour of the translation-based approach for different types of formulae. I trust that the authors will do this in their future work.

What the Related work is missing, is probably "AutoHyper: Explicit-State Model Checking for HyperLTL", Raven Beutner and Bernd Finkbeiner, TACAS, Lecture Notes in Computer Science, 145-163 (2023), where the authors evaluate hyperproperties by applying language inclusion checks instead of explicit automaton complementation, and that can apply to arbitrary HyperLTL formulae as well as the current paper's approach.

Overall a good "Preliminary Work" short paper for ICAPS. We are waiting for a more complete paper.

[1] Tzu-Han, Hsu., C'esar, S'anchez., Sarai, Sheinvald., Borzoo, Bonakdarpour. (2023). Efficient Loop Conditions for Bounded Model Checking Hyperproperties.  66-84. doi: 10.48550/arXiv.2301.06209

# Minor considerations
- Sec.1 It is not clear what form has the output o in the example (it's a plan?)
- Def. 1 For each i \in I -> for each i \in I
- Sec. 6 I prefer the term "LTL fragment" rather than "LTL-like body" for a formula
- Sec. 6 I would rather incluse the alphabet Sigma in the definition of A, i.e. A = (Q, q0, rho, F, Sigma).
- Def. 2 The "forall j" of l_j \in L is not necessary
- Def. 2 Better to specify that j\in [1,n] when writing d_j \in D.
- It would be helpful for the reader to include an example of a formula in order to illustrate a TS.

---

> ### Author Rebuttal · Authors · 2024-01-27
>
> We thank the reviewer for their positive review and very valuable feedback.
> We will carefully take the reviewer's feedback into account when preparing a final version.
> In our evaluation, we (for space reasons) focus on the parity-game-based approach, as (1) it is closest to our novel planning-based approach, and (2) it best demonstrates the performance impact of the factored representation within our planning encodings.
>
> **Questions:**
>
> 1) A DFA differs from a Büchi automaton in that we only need to visit an accepting state _once_ (not infinitely many times).
> As we argued at the end of Section 6.2, we could extend our planning-based encoding to full HyperLTL by adding _temporal objectives_.
> For example, in the case of a Büchi automaton, a plan would need to visit a goal state infinitely often.
> This goes beyond the standard reachability conditions studied in many FOND planning settings (Cimatti et al. 2003).
>
> 2) In our QDec-POMDP, the planner selects a direction for all existentially quantified traces, and, based on this action, the model can transition to multiple states (i.e., all states that can be reached by non-deterministically picking directions for universally quantified traces).
> In the QDec-POMDP semantics (cf. Section 4), we then pick _one_ of these potential successor states (i.e., we select a _single_ outcome).
> Each agent observes which state we picked based on its observation function (in a FOND setting, it observes the entire state) and can select a new action.
> A strong plan (in Cimatti's sense) must reach a goal state in _all_ possible outcome sequences (similar to a multi-player game).
> As noted by the reviewer, constructing belief states is one possible method for _solving_ a QDec-POMDP, but it is not necessary for FOND instances where we deal with full observations.
> In our paper, we leave the question of _how_ to solve the FOND instance to off-the-shelf tools.
>
> 3) We consider our QDec-POMDP translation as a proxy that allows us to leverage planning tools for hyperproperty verification.
> In terms of expressiveness, it is hard to characterize in which situations the translation yields a QDec-POMPD that admits a plan.
> For $\forall^*\exists^* G (...)$ properties, we conjecture that a plan exists iff there exists a simulation relation (see, e.g., [1], (Hsu et al. TACAS 2023)).
> For properties beyond the $\forall^*\exists^*$ fragment, a precise characterization is challenging due to the presence of _partial_ observations.

---

### Meta-Review · Area_Chair_27LZ · 2024-02-06

**Recommendation:** Accept (Poster)
**Confidence:** 5

**Metareview:**

The reviewers agree that the paper presents an interesting contribution, relevant to ICAPS, and overall of good quality. Some minor issues were raised, which appear to be easily addressable in the final version, if the paper is accepted. In particular, I recommend the authors to cover also the additional related work mentioned by in the reviews. Additional suggestions are intended for a future extended version of the work, but the authors may try and account for some of them briefly in the present submission, such as the comparison with model checking.

**Ethical Considerations:**

(1) Not Applicable: The paper does not have any ethical considerations to address